# Distinctive Neuroanatomic Regions Involved in Cocaine-Induced Behavioral Sensitization in Mice

**DOI:** 10.3390/biomedicines11020383

**Published:** 2023-01-27

**Authors:** Renan dos Santos-Baldaia, Raphael Wuo-Silva, Viviam Sanabria, Marilia A. Baldaia, Thais S. Yokoyama, Antonio Augusto Coppi, André W. Hollais, Eduardo A. V. Marinho, Alexandre J. Oliveira-Lima, Beatriz M. Longo

**Affiliations:** 1Laboratory of Neurophysiology, Department of Physiology, Universidade Federal de São Paulo, São Paulo CEP 04023-066, Brazil; 2Department of Pharmacology, Universidade Federal de São Paulo, São Paulo CEP 04023-066, Brazil; 3Department of Neurology and Neurosurgery, Universidade Federal de São Paulo, São Paulo CEP 04023-066, Brazil; 4Bristol Veterinary School, University of Bristol, Bristol BS40 5DU, UK; 5Department of Physiological Sciences, Universidade Federal do Espírito Santo, Espírito Santo CEP 29075-910, Brazil; 6Department of Health Sciences, Universidade Estadual de Santa Cruz, Ilhéus, Bahia CEP 45662-900, Brazil

**Keywords:** cocaine, behavioral sensitization, induction phase, expression phase, brain structures, c-Fos, dopamine

## Abstract

The present study aimed to characterize the phenomenon of behavioral sensitization to cocaine and to identify neuroanatomical structures involved in the induction and expression phases of this phenomenon. For this, in experiment 1 (induction phase), mice were treated with saline or cocaine every second day for 15 days (conditioning period), in the open-field or in their home-cages. In experiment 2 (expression phase), the same protocol was followed, except that after the conditioning period the animals were not manipulated for 10 days, and after this interval, animals were challenged with cocaine. Neuroanatomical structures involved in the induction and expression phases were identified by stereological quantification of c-Fos staining in the dorsomedial prefrontal cortex (dmPFC), nucleus accumbens core (NAc core and shell (NAc shell), basolateral amygdala (BLA), and ventral tegmental area (VTA). Neuroanatomical analysis indicated that in the induction phase, cocaine-conditioned animals had higher expression of c-Fos in the dmPFC, NAc core, BLA, and VTA, whereas in the expression phase, almost all areas had higher expression except for the VTA. Therefore, environmental context plays a major role in the induction and expression of behavioral sensitization, although not all structures that compose the mesolimbic system contribute to this phenomenon.

## 1. Introduction

Addiction is a neurobiological disorder in which repeated drug use reorganizes the neural pathways that mediate reward and adaptive behaviors, causing neuroplastic changes. Such changes are associated with compulsive drug-seeking behaviors, the inability to control such behaviors, and the continued use of substances [1,2]. In addition to the negative consequences, there is a vulnerability to relapse, even after prolonged periods of abstinence. Moreover, addiction leads to the devaluation of natural rewards such as food, home, and sexual behavior, which are vital to survival [1,2,3].

At the level of neuronal circuitry, it has been shown that drug abuse causes changes in several brain areas, especially those involved in the reward system, such as the striatum, the basal ganglia, the limbic system, and the prefrontal cortex [4,5]. Likewise, most abused drugs stimulate the mesoaccumbens dopaminergic pathway [6].

Many of the findings related to the neurobiological mechanisms involved in the addiction process were only possible due to the use of animal models. Among the most well-known models, the behavioral sensitization phenomenon stands out, in which the repeated administration of psychostimulant drugs, such as cocaine, promotes a progressive and long-lasting increase in the activity of the mesocorticolimbic dopaminergic system, leading to a corresponding increase in locomotor-stimulating effect in rodents [7,8,9]. This phenomenon has been used to study the neurochemical mechanisms involved in dopaminergic mesocorticolimbic neuroplasticity, which is related to the effects of reinforcement, incentive salience, and craving induced by drugs with abuse potential in humans [9,10]. 

The behavioral sensitization phenomenon could be divided into two phases: the induction phase (or development) that corresponds to the behavioral responses displayed by the animals during repeated treatment with the drug, and the expression phase that corresponds to the behavioral responses elicited by a challenge injection of the drug, administered during withdrawal from the repeated treatment [11,12,13]. 

It is essential to highlight that studies have shown that behavioral sensitization induced by psychostimulants, such as amphetamine and cocaine, can be potentiated by environmental cues previously paired with the effects of these drugs [14,15,16]. Thus, in this model, only the presentation of environmental cues, in the absence of drugs, can evoke enhanced locomotor activity in rodents, called conditioned locomotion [17,18]. This phenomenon would be analogous to what occurs in drug-dependent humans, in which clues related to its use resemble its effects, generating a compulsive desire for the drug [18,19]. In this context, it has been demonstrated that the same brain structures can be related to both the behavioral sensitization and the environmental conditioning process, such as a greater activation of dopaminergic transmission in structures such as the NAc and VTA [20,21]. 

Regarding the increase in locomotor stimulant response, it must be considered that different neural circuits underlie the two distinct phases of the behavioral sensitization phenomenon [22]. Hence, the present study aimed to characterize the phenomenon of locomotor sensitization to cocaine through behavioral effects, and to identify possible neuroanatomical structures involved in this phenomenon through the stereological quantification of the expression of the c-Fos protein. We aimed to determine which structure activations correspond to the induction and expression phases when cocaine is administrated in a paired or not environmental context.

## 2. Materials and Methods

### 2.1. Animals

Ninety-one female 3-month-old Swiss EPM-M2 mice (weighing between 25–35 g; from the CEDEME- Center for the Development of Animal Models at Universidade Federal de São Paulo (UNIFESP) were housed at room temperature (21 ± 2 °C), 12:12 light/dark cycle (lights on at 7:00 a.m.), with free access to food and water. All measures were taken to minimize pain and discomfort throughout the study. All animal handling and experimental procedures complied with the guidelines for animal care and use of laboratory animals and were approved by the Board for Ethics in the Use of Animals (CEUA, Comissão de Ética no Uso de Animais) and the institutional ethics committee of the UNIFESP, protocol nº 6922070715. Female mice were used due to their high susceptibility to drugs, and to their robust and rapid behavioral response [23,24,25,26]. Moreover, the use of female mice positively contributes to a more ethical laboratory housing practice [27,28,29].

### 2.2. Groups

The distribution of the groups is represented in Table 1.

### 2.3. Drugs

Cocaine-HCl (Sigma®) was diluted in 0.9% saline solution. Both cocaine (10 mg/kg body weight) and saline solution were administered intraperitoneally on the odd days (1, 3, 5, 7, 9, 11, 13, and 15) of the induction phase at the same schedule each day. Cocaine (10 mg/kg) was also administrated alone during the cocaine challenge (day 26). The selected dose of cocaine used in the present study was based on previous studies conducted by our group [17,28,30].

### 2.4. Open-Field Locomotor Activity 

Locomotor activity was measured in the open-field apparatus. The apparatus consisted of an open-top cylinder with a circular plastic wall (40 cm in diameter and 50 cm high), with a floor divided into 19 similar trapezoid-shape sectors of approximately 67.51 cm^2^ each, delimited by three concentric circles of different radii (8, 14, and 20 cm) intersected by radial line segments [31]. Animals were placed in the behavioral testing room at least 1 h before beginning the behavioral tasks to minimize possible handling stress from moving animal cages between the vivarium and the testing room. While in the testing room, animals were exposed to normal light mimicking the lighting conditions of the vivarium. Hand-operated counters were used by an observer who was blind to the treatment to score total locomotion frequencies (i.e., the total number of entries into any sector with the four paws) during the 10 min sessions at the same clock time. For each animal, the number of entries was summed, which in turn generated a mean value for each group, and for statistical analysis, all animals of all groups were considered. After each animal test, the open-field apparatus was cleaned with a solution of alcohol–water 5%.

### 2.5. Experimental Procedures

#### 2.5.1. Experiment 1: Characterization of the Induction Phase of Cocaine-Induced Hyperlocomotion

Forty-six female mice were habituated in the open-field for 10 min for three consecutive days before the conditioning period. Basal locomotor activity was measured on the third day of the habituation period. After the 3-day habituation, animals were divided into three groups: Sal-Sal, Coc-Sal, and Sal-Coc (N = 15–16 per group). The animals in the Sal-Sal group received one saline solution injection via intraperitoneal (i.p.), and after 5 min, they were exposed to the open-field for 10 min. Then, 2 h later, the animals were injected again with saline solution in their respective home-cage. Animals in the Coc-Sal group received one injection of cocaine (10 mg/kg, i.p.); 5 min later they were exposed to the open-field for 10 min. After 2 h, one saline injection was administered in their respective home-cages. Animals in the Sal-Coc group received one saline injection, and after 5 min, they were exposed to the open-field for 10 min. After 2 h, the animals received one injection of cocaine (10 mg/kg, i.p.) in their respective home-cage.

Injections of saline and cocaine were applied every second day for 15 days on the odd days (1, 3, 5, 7, 9, 11, 13, and 15). The locomotor activity was evaluated on days 1 and 15 of the induction phase. During the alternate nontreatment days (2, 4, 6, 8, 10, 12, and 14), mice were not handled, and stayed in their home-cages. Animals in the Sal-Coc group received cocaine injections outside the environmental context, and animals in the Coc-Sal group received cocaine injections in a paired way to the environmental context; however, both groups had the same pharmacological treatment (see the distribution of the groups and treatment received in Table 1 and experimental design in Figure 1).

#### 2.5.2. Experiment 2: Characterization of the Expression Phase of Cocaine-Induced Hyperlocomotion

Forty-five female mice were habituated, and locomotor activity was measured as described above (see explanation in Section 2.5.1). After the 3-day habituation phase, animals were divided into three groups (N = 15 per group) Sal-Sal-Coc, Coc-Sal-Coc, and Sal-Coc-Coc, and the same protocol as Section 2.5.1 was followed. After the fifteen-day conditioning period, the animals stayed for 10 days in their home-cages without experimental manipulations. At the end of the no-handling period of 10 days, all animals of the three groups received 10 mg/kg, i.p. of cocaine (cocaine challenge), and after 5 min they were exposed to the open-field for 10 min (see Table 1 and Figure 1 for clarification). 

### 2.6. Tissue Preparation 

Ninety minutes after the last behavioral test, animals were deeply anesthetized and perfused through the heart with 100 mL of 0.1 M PBS solution (phosphate-buffered solution; 5.52 g of NaH2PO4 + 21.88 g of Na2HPO4), followed by 100 mL of 4% formaldehyde (diluted in PBS solution). The brains were removed and post-fixed in a 4% formaldehyde solution followed by a hypertonic solution of sucrose 30% (diluted in PBS) for cryoprotection. The brains remained in this solution until they showed signs of dehydration, a procedure that usually occurs within 24 h. Immediately after this period, the brains were dried and frozen at -80 ºC. Subsequently, the brains were sectioned in a cryostat (Leica CM1850) in coronal brain sections (50 µm thick) at intervals of three slices. For every three sections, one was selected (for immunohistochemistry processing), and two were discarded. Then, the sections were stored in an antifreeze solution (300 g of sucrose, 500 mL of PBS solution, and 300 mL of ethylene glycol) at -20 ºC until immunohistochemical processing.

### 2.7. c-Fos Protein Expression

Free-floating sections were washed five times with PBS solution to remove all antifreeze solution and incubated in a blocking solution (90 mL of 0.1 M PBS, 250 µL of Triton X-100, 40 µL of normal goat serum) for 30 min at room temperature under constant stirring. After this period, the sections were incubated overnight with a polyclonal primary antibody anti-c-Fos made in rabbit (at a dilution of 1:3000, Calbiochem, Merck KGaA, Darmstadt, Germany) diluted in a blocking solution.

The following day, after washing in PBS, the sections were incubated in goat anti-rabbit secondary antibody (1:200 dilution, Vectastain Vector, Burlingame, CA, USA) diluted in a blocking solution for 2 h, followed by incubation in the ABC kit solutions (Vectashield, Vector, Burlingame, CA, USA) for 1.5 hours. The sections were stained with diaminobenzidine (DAB, Sigma-Aldrich Corporation, St. Louis, USA), mounted on slides, and sealed with coverslips.

### 2.8. Stereological Analysis

Stereological analysis was performed using a microscope (Nikon Eclipse 80i) with a high-resolution CCD video camera coupled to a computer installed with the StereoInvestigator software (MicroBrightField, version 9, Williston, VT, USA).

The analysis of the sections started systematically and uniformly randomly (SURS) in each of the regions. For every three slices, one was chosen; thus, all regions had an interval of three slices (or 150 µm). In addition, slices that contained the dorsomedial prefrontal cortex (dmPFC), the nucleus accumbens core and shell (NAc core and NAc shell, respectively), the amygdala basolateral (BLA), and the ventral tegmental area (VTA) were selected. Contour delineations of all analyzed structures (right and left hemisphere) were made according to the atlas Paxinos and Franklin (2012) [32] using an X2 lens. The counts for c-Fos corresponded to the bregma plane +2.46 – +1.34 (dmPFC), +1.70 – +0.86 (NAc core and shell) -0.58 – -1.82 (BLA), and -2.92 – -3.88 (VTA). All analysis was performed using an X100 with an oil immersion lens and applying the optical fractionator method.

A pilot study was carried out for each region analyzed to determine the stereological parameters (Table 2). 

The optical fractionator method was used to estimate the total number of cells in each brain structure. The estimation precision is determined by the coefficient of error (CE); this value expresses the intraindividual variation and the error attributed to the stereological method. As a general rule, CE should be at most 10%. Counting was performed blindly to treatments, and visualization of counts was accessed after the analysis was complete. 

### 2.9. Statistical Analysis

Behavioral experiments and stereological data were first analyzed for normality by the Shapiro–Wilk test. Then, the one-way ANOVA (analysis of variance) test was performed, followed by Duncan’s post hoc test to detect the effects of the treatments (saline or cocaine). In addition, the GLM repeated measures test was performed for comparisons within the same experimental group at different times. A *p*-value less than 0.05 was considered to be a statistically significant difference. Statistical analyzes were performed using GraphPad Prism 5 software (San Diego, CA, USA and PASW Statistics 18 software (SPSS Inc., Chicago, IL, USA).

## 3. Results

### 3.1. Experiment 1

#### 3.1.1. Characterization of the Induction Phase of Cocaine-Induced Hyperlocomotion

All animals were habituated during three consecutive days in the open-field to eliminate the novelty effect, and had their locomotor activity quantified on the third day. There was no significant difference among the groups (Sal-Sal, Coc-Sal, and Sal-Coc). However, after analyzing the first day of cocaine treatment, it was observed that the animals that received cocaine paired with the open-field (Coc-Sal group) presented a significant increase in locomotor activity when compared to the control group (Sal-Sal) (F (2,36) = 3.38; *p* < 0.022; Figure 2). 

It is essential to highlight that the animals from the Sal-Coc group received cocaine after saline injection, which means that these animals were not exposed to the open-field under the effect of cocaine. However, they have the same pharmacological history as the Coc-Sal group. 

Moreover, on the last day of conditioning, animals from the Coc-Sal group that received cocaine for 15 intermittent days and were exposed to the open-field presented a significant increase in locomotor activity when compared to the Sal-Sal group (F (2,36) = 5.70; *p* < 0.001), and when compared to themselves on the first day of injection of cocaine (F (2,36) = 3.51; *p* < 0.003; Figure 2), characterizing the development of behavioral sensitization.

#### 3.1.2. Stereological Evaluations of the Expression Phase of Cocaine-Induced Hyperlo-comotion

##### dmPFC

The effective number of c-Fos cells counted in this region varied between 286 and 347, with an average number of 303 cells. CE was within the recommended range, thus varying between 0.04 and 0.09, with an average of 0.061. When analyzing the groups, there was a significantly higher expression of c-Fos protein in the dmPFC in the Coc-Sal group when compared to the Sal-Sal control group (F (2,15) = 3.29; *p* < 0.039; Figure 3A).

##### NAc Core

The effective number of c-Fos in this region varied between 133 and 349, with an average number of 231 cells. CE was within the recommended range, thus varying between 0.03 and 0.08, with an average of 0.062. When analyzing the groups, there was a significantly higher expression of c-Fos protein in the NAc core of the Coc-Sal group when compared to Sal-Coc and Sal-Sal group (F (2,15) = 12.83; *p* < 0.002; Figure 3B).

##### NAc Shell

The effective number of c-Fos in this region varied between 255 and 465, with an average number of 340 cells. CE was within the recommended range, thus varying between 0.06 and 0.08, with an average of 0.066. When analyzing the groups, there were no significant differences between groups (F (2,15) = 3.10; *p* = 0.390).

##### BLA

The effective number of c-Fos in this region varied between 266 and 471, with an average number of 304 cells. CE was within the recommended range, thus varying between 0.03 and 0.08, with an average of 0.061. When analyzing the groups, there was a significantly higher expression of c-Fos protein in the BLA of the Coc-Sal group when compared to the Sal-Sal control group (F (2,15) = 2.42; *p* < 0.037; Figure 3C). 

##### VTA

The effective number of c-Fos in this region varied between 99 and 224, with an average number of 144 cells. CE was within the recommended range, thus varying between 0.07 and 0.15, with an average of 0.011. When analyzing the groups, there was a significantly higher expression of c-Fos protein in the VTA of the Coc-Sal group when compared to the Sal-Sal control group (F (2,15) = 1.69; *p* < 0.004; Figure 3D). Moreover, there was a significant decrease in the expression of c-Fos in the Sal-Coc group compared to the Coc-Sal group (F (2,15) = 2.77; *p* < 0.0350).

### 3.2. Experiment 2

#### 3.2.1. Characterization of the Expression Phase of Cocaine-Induced Hyperlocomotion

All animals were habituated for three consecutive days in the open-field, and their general locomotion was quantified on the third day with no significant differences among groups (Sal-Sal-Coc, Coc-Sal-Coc, and Sal-Coc-Coc). On the first day of cocaine treatment, the animals that received cocaine and were exposed to the open-field showed hyperlocomotion when compared to the control group (F (2,35) = 5.84; *p* < 0.006), as well as on the last day of cocaine and when compared to themselves on the first day of cocaine (F (2,35) = 30.87; *p* < 0.0001).

After the last day of cocaine, the animals remained abstinent from the drug for 10 days; after this period, they were challenged with cocaine. Animals with a history of cocaine condition in the open-field (Coc-Sal-Coc) presented hyperlocomotion when compared to the control group (F (2,35) = 4.06; *p* < 0.025; Figure 4), demonstrating that the animals expressed the phenomenon of behavioral sensitization.

#### 3.2.2. Stereological Evaluations of the Expression Phase of Cocaine-Induced Hyperlocomotion

##### dmPFC

The effective number of c-Fos in this region varied between 198 and 498, with an average number of 301 cells. CE was within the recommended range, thus varying between 0.04 and 0.08, with an average of 0.062. When analyzing the groups, there was a significantly higher expression of c-Fos protein in the dmPFC of the Coc-Sal-Coc group when compared to the Sal-Sal-Coc and Sal-Coc-Coc groups (F (2,15) = 8.73; *p* < 0.001; Figure 5A).

##### NAc Core

The effective number of c-Fos in this region varied between 133 and 398, with an average number of 222 cells. CE was within the recommended range, thus varying between 0.07 and 0.12, with an average of 0.10. When analyzing the groups, there was a significantly higher expression of c-Fos protein in the NAc core of the Coc-Sal-Coc group when compared to the Sal-Sal-Coc control group (F (2,9) = 1.87; *p* < 0.003; Figure 5B).

##### NAc Shell

The effective number of c-Fos in this region varied between 303 and 478, with an average number of 391 cells. CE was within the recommended range, thus varying between 0.04 and 0.07, with an average of 0.06. When analyzing the groups, there was a significantly higher expression of c-Fos protein in the NAc shell of the Coc-Sal-Coc group when compared to the Sal-Sal-Coc control group (F (2,15) = 22.08; *p* < 0.003; Figure 5C). 

##### BLA

The effective number of c-Fos in this region varied between 222 and 524, with an average number of 309 cells. CE was within the recommended range, thus varying between 0.03 and 0.09, with an average of 0.066. When analyzing the groups, there was a significantly higher expression of c-Fos protein in the BLA of the Coc-Sal-Coc group when compared to Sal-Coc-Coc and Sal-Sal-Coc groups (F (2,15) = 2.87; *p* < 0.002; Figure 5D). 

##### VTA

The effective number of c-Fos in this structure region is between 112 and 398, with an average number of 255 cells. CE was within the recommended range, thus varying between 0.07 and 0.10, with an average of 0.088. When analyzing the groups, there were no significant differences between groups. 

## 4. Discussion 

The present study characterizes the behavioral sensitization phenomenon to cocaine and shows neuroanatomical structures involved in both phases (induction and expression) of this phenomenon when cocaine is administered in a paired way or not to the environmental context. Our results showed that, in the induction phase of cocaine conditioning (experiment 1), the animals that received cocaine paired with the open-field (Coc-Sal group) presented a significant increase in locomotor activity compared to the control group (Sal-Sal), which characterizes the development of behavioral sensitization. Moreover, in the induction phase, the expression of the c-Fos protein was increased in the dmPFC, NAc core, BLA, and VTA structures in the Coc-Sal group that were first exposed to cocaine compared to the control group (Table 3). 

Regarding the characterization of the expression phase of cocaine (experiment 2), the animals that received the drug and were exposed to the open-field (Coc-Sal-Coc group) showed hyperlocomotion on the first and last day of cocaine compared to the control group. This result indicates that the animals expressed the phenomenon of behavioral sensitization. However, the cocaine challenge did not evoke locomotor sensitization at the same magnitude of locomotion observed in the induction phase, which may be explained by the ceiling effect [33,34], corroborating what was described for ethanol and amphetamine [35,36]. 

When the expression of c-Fos was analyzed, a significant increase was observed in dmPFC, in both areas of the nucleus accumbens (NAc core and shell), and BLA in the Coc-Sal-Coc group when it was compared to the control group (Table 3).

Curiously, in animals that received injections outside the environmental context in the induction phase, only the VTA increased c-Fos expression in the Sal-Coc group. In contrast, in the expression phase, only the dmPFC increased c-Fos expression in the Sal-Coc-Coc group. Therefore, we can assume that these two regions differentiate the two phases in these groups. 

The importance of the mesolimbic dopaminergic pathway in behavioral sensitization was demonstrated by the greater expression of c-Fos in the NAc core and the VTA in the animals of the Coc-Sal group during the induction phase of cocaine. To corroborate these data, pharmacological lesions in rats of both VTA and NAc structures produce hypoactivity and complete blockade of locomotor-stimulating effects of another psychostimulant, amphetamine [37].

Steketee and Kalivas (2011) suggested that the induction phase of the behavioral sensitization phenomenon is due to the action of the drugs of abuse on the cell bodies of dopaminergic neurons located in the VTA, while the expression of this phenomenon is a consequence of an increase in the release of dopamine (DA) and an increase in postsynaptic responsiveness to DA in the NAc [21]. According to Ford (2014), the different stages of the behavioral sensitization process probably involve different types of neuronal alterations in the mesoaccumbens dopaminergic system. Thus, the induction of this process seems to involve a decreased sensitivity of D2-type dopaminergic autoreceptors. In contrast, the expression of behavioral sensitizations appears to be due to an increase in DA release [38].

Interestingly, in our study, we observed that in the induction phase the group that received cocaine paired or not to the environmental context had an increase in c-Fos expression in the VTA, although, in the expression phase, the same increase outside and paired to the environmental context occurred in the dmPFC region. These data suggest that cocaine stimulates VTA neurons during the induction phase. However, in the expression phase, this would be related to the stimulation of neurons in the PFC, perhaps not directly to the action of cocaine, but rather to an increase in dopamine in this area. 

In the expression phase of cocaine, there was a greater expression of c-Fos in the NAc core in the Coc-Sal-Coc group, not as robust as in the induction phase, however, different from the control group, while in VTA, unlike in the induction phase, there was no difference among groups in the expression of c-Fos in this structure. Repeated amphetamine injections in the VTA, but not in the NAc, produce sensitization to the locomotor response induced by a subsequent peripheral injection [39]. These results suggest that while psychostimulants promote the induction of behavioral sensitization to the locomotor response, it also depends on the action of the drug in the VTA; the expression of the phenomenon seems to be related to its activity in the dopaminergic ending located in the NAc [40].

Another system that is important in the behavioral sensitization process is the glutamatergic system. In the last two decades, it has been shown that excitatory glutamatergic transmission over the VTA is necessary for the development of sensitization to drugs of abuse [41,42,43]. Ferrario and colleagues (2010) mentioned that glutamatergic connections from various regions of the limbic system and prefrontal cortex on NAc neurons are fundamental for the expression of behavioral sensitization. Glutamate excites medium spinal neurons in the NAc through AMPA-like receptors [44]. After drug withdrawal, changes in the expression of these receptors occur, which influence the excitability of neurons from the NAc and the drug-seeking behaviors mediated by NAc [45].

In our study, no significant difference in the expression of c-Fos among the groups in the NAc shell was observed in the induction phase; however, there was a greater expression of c-Fos in the NAc shell of animals paired with cocaine (Coc-Sal-Coc) in the expression phase, demonstrating that the NAc shell subregion has no participation in the induction phase but has great importance in the expression phase of behavioral sensitization due to the direct action of cocaine.

To verify NAc shell relevance, Todtenkopf and colleagues (2002) verified the expression of c-Fos in the NAc shell and its subdivision in animals pretreated with cocaine and challenged with the same drug 2 or 14 days after the last day of repeated drug administration. The authors verified that when the animals were challenged 2 days later, there was a lower expression of c-Fos in the ventrolateral subdivision of the NAc shell, while no difference was observed in the other subdivisions [46]. Nevertheless, 14 days later, there was a greater expression of c-Fos in the intermediate subdivision of the NAc shell, demonstrating that different regions in the NAc shell have different functions when challenged with cocaine at different times [45].

Likewise, the NAc shell plays an essential role in the development of associative processes between drugs of abuse and environmental context [46]. Bossert and colleagues (2007) examined the effect of SCH 23390 injections (dopamine D1-family receptor antagonists) into the medial and lateral shell and core on discrete-cue-induced reinstatement of heroin-seeking. Using the self-administration model, these authors verified a decrease in the conditioned response phenomenon in the NAc shell (but not in the NAc core) [46].

Moreover, previous studies indicated that in the expression phase in a cocaine-paired environment, the structures dmPFC, NAc core and shell, and BLA exhibit neuronal activation of cocaine in conjunction with cocaine-seeking behavior [47,48]. Other limbic regions, such as the amygdala, through glutamatergic projections to the VTA, NAc, and hippocampus, seem to modulate the activity of the mesoaccumbens pathway, contributing to the development and expression of behavioral sensitization [49,50]. Emphasizing the integration between the BLA, NAc, and the neurotransmitter glutamate, Kalivas (2002) suggested that the recruitment of glutamatergic cortical regions by environmental stimuli previously associated with the use of the drug, simultaneously with the behavioral manifestation of dependence, favors the idea of a transition from behaviors primarily dependent on dopaminergic transmission (elicited by acute drug administration) to behaviors primarily dependent on glutamatergic transmission (produced by drug-paired environmental stimuli) [50]. Such transition would occur physiologically during adaptation to natural stimulants such as new environments, food, and sex [51]. Considering the above scenario, more recent evidence suggests that the BLA sends projections to the NAc, modulating motivated behaviors.

Ambroggi and colleagues (2008) demonstrated that reward-seeking behavior requires both the activation of glutamatergic neurons in BLA towards the NAc and the activation of dopamine D1 receptors in the NAc itself [52]. Likewise, Fuchs and collaborators (2004) mentioned that the dmPFC structure is fundamental for cocaine-primed reinstatement of cocaine-seeking behavior [48].

In fact, in both phases, the development (experiment 1) and the expression (experiment 2) of behavioral sensitization, there was a greater expression of c-Fos in the BLA in animals with a history of cocaine associated with the environmental context, which may be explained by the critical role that the amygdala plays in the reward circuit, mainly associating pharmacological and environmental effects [53]. 

The repeated use of a drug of abuse, more specifically cocaine, leads to associative learning, so the individual associates the euphoric effect promoted by the drug with the environment where it is consumed [9]. As a result, a reinforcing system is established, meaning that two stimuli will be associated with choice or compulsive desire for drug use [54]. 

Marinelli and colleagues (2007) carried out an experiment using an animal model of self-administrated alcohol and demonstrated the importance of the BLA in developing an associative process between the effects of drugs and the environmental context [55]. These authors verified a higher c-Fos mRNA expression in rats that self-administered alcohol in a known context. It has been shown that most drugs of abuse share neural substrates [55]. In this sense, the results obtained by the present work regarding the expression of c-Fos in the BLA corroborate the results found by Marinelli and colleagues [55]. In addition, injuries to the BLA of rats prevented self-administration in a conditioned context; however, the authors did not alter cocaine self-administration in a different environment. Meil and See (1997) suggested that this structure is more related to the conditioned properties of cocaine and not to the immediate primary reinforcement [56].

Furthermore, Volkow and Fowler (2000) suggested that the amygdala and NAc have an essential role, mainly in the initial effects promoted by drugs of abuse [57].

Other structures that have a relevant role in the modulation of behavioral sensitivity to cocaine are the acid-sensing ion channels (ASICs) which seem to participate in the drug-regulated synaptic plasticity. Jiang and colleagues (2013) examined the role of ASIC1a and ASIC2 in regulating behavioral sensitivity to psychostimulant cocaine by utilizing knockout mice. They observed that knockout mice displayed decreased cocaine sensitization to chronic and acute cocaine administration as compared to controls [58].

In a nutshell, the behavioral sensitization to cocaine proved to be a behavioral phenomenon with a conspicuous and lasting manifestation under our experimental conditions. The dmPFC region has been shown to play an essential role in the induction and expression phase of behavioral sensitization, mainly in the paired administration of cocaine. The NAc core region is activated in both phases (induction and expression) of cocaine-induced locomotor sensitization. However, such activation is more robust in the sensitization development phase, while the NAc shell was explicitly activated in the expression phase of sensitization. In contrast, the BLA region is activated during the development of behavioral sensitization and, mainly, during the expression of locomotor sensitization to cocaine. The VTA was explicitly activated during the development, mainly in the paired administration of cocaine, although it did not change during the expression of behavioral sensitization to cocaine. 

## 5. Conclusions

Neuroanatomical studies related to the positive reinforcing properties of drugs of abuse and their association with specific environmental contexts, which use design-based stereological methods, are scarce in the scientific literature. Complementary investigations in other regions involved in the dependence circuitry become necessary to shed light on our understanding of the mechanisms and regions related to the phenomenon of behavioral sensitization. The results of the present study strengthen and complement those findings by showing the participation of different structures in the induction and/or expression of behavioral sensitization to cocaine in a paired or not context.

## 6. Study limitations

Some limitations of our research were that we chose not to determine the estrous cycle of females due to the stress generated by the vaginal smear. Instead of monitoring the cycle, we used a heterogeneous population of female mice, which generated robust statistical data indicating significant differences based on heterogeneity. In addition, locomotion activity was measured by researchers that were blind to the treatment, although automatic software would be ideal. Other limitations concern the sensibility of c-Fos antibody to different stimuli and difficulties in standardizing stereological parameters. All limitations identified in this study should be addressed in future studies.

## Figures and Tables

**Figure 1 biomedicines-11-00383-f001:**
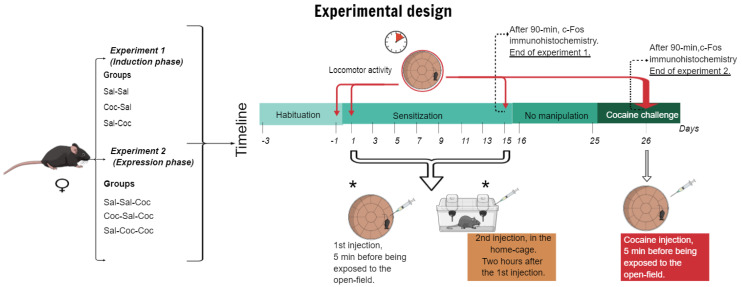
Experimental designs of the induction (experiment 1) and expression (experiment 2) phase of cocaine-induced hyperlocomotion with a timeline. Groups were distributed according to the experiment. (*) Represents that the first and second injection during the sensitization period was administered concerning the group treatment. Both injections were applied on the same day. During the sensitization period, the injections were applied on the odd days (1, 3, 5, 7, 9, 11, 13, and 15). The first injection was applied in the open-field and two hours later in the animal´s home-cage. On the even days (2, 4, 6, 8, 10, 12, and 14), animals were not handled.

**Figure 2 biomedicines-11-00383-f002:**
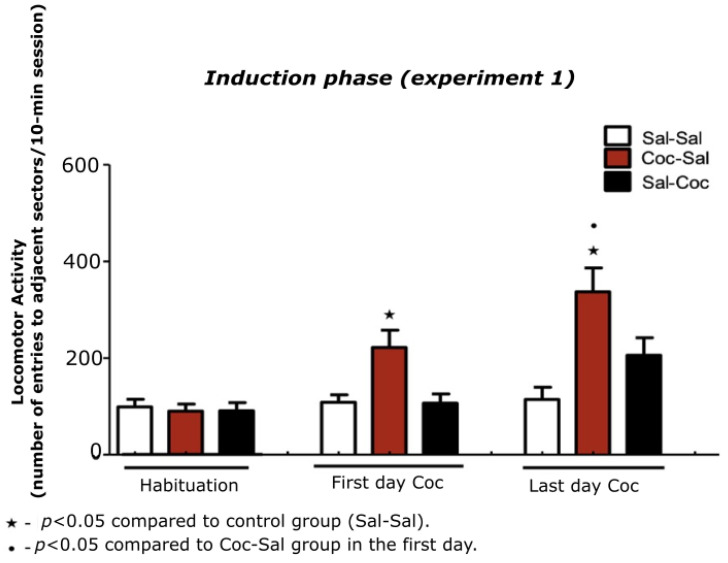
Total locomotion of the animals in the induction phase of cocaine-induced behavioral sensitization.

**Figure 3 biomedicines-11-00383-f003:**
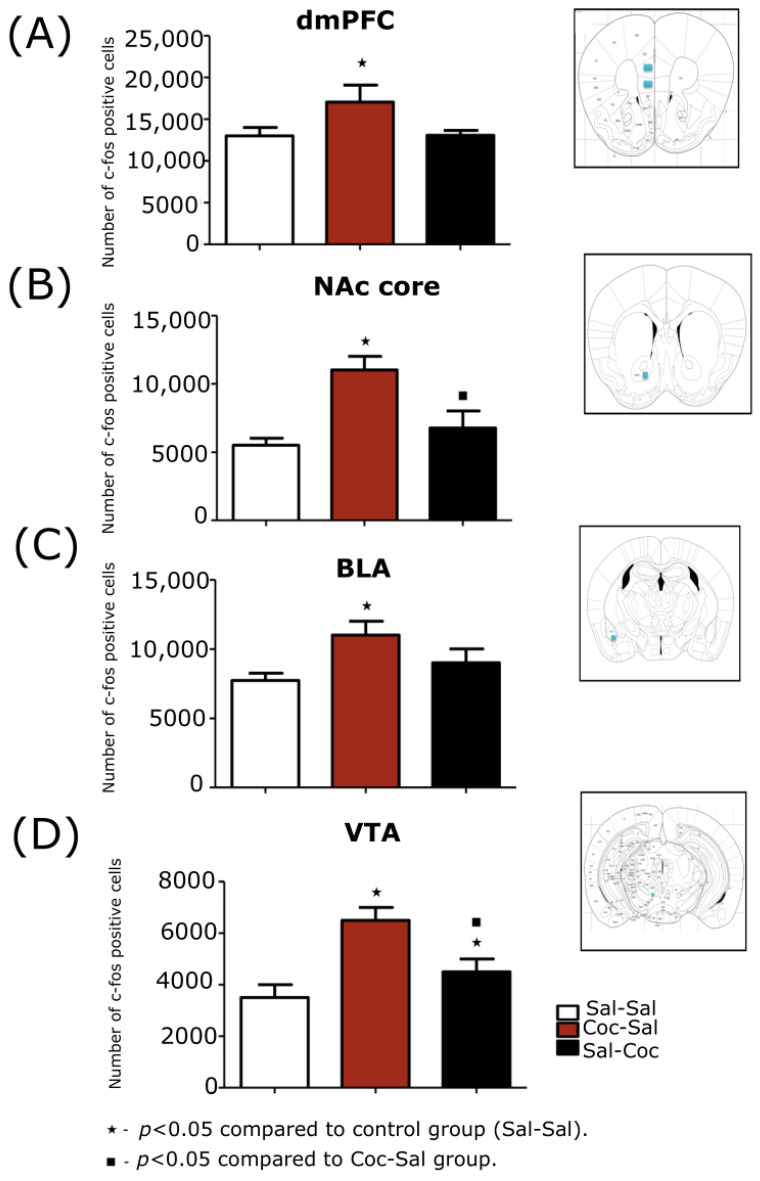
Stereological analysis of the expression c-Fos protein in animals treated with cocaine, administered in a paired way (Coc-Sal) or not (Sal-Coc) to the environmental context in the induction phase in the following structures: (**A**) The dorsomedial prefrontal cortex (dmPFC) measurements were performed at 1.95 mm of bregma, a representative area of quantification is observed by the blue rectangles on the right. (**B**) The nucleus accumbens core (NAc core) measurements were performed at 1.18 mm of bregma, a representative area of quantification is observed by the blue rectangle on the right. (**C**) The basolateral amygdala (BLA) measurements were performed at −1.22 mm of bregma, a representative area of quantification is observed by the blue rectangle on the right. (**D**) The ventral tegmental area (VTA) measurements were performed at −2.92 mm of bregma, a representative area of quantification is observed by the blue rectangle on the right. The number of c-Fos positive cells is expressed as mean ± SEM. The one-way ANOVA test was performed, followed by the Duncan post hoc test.

**Figure 4 biomedicines-11-00383-f004:**
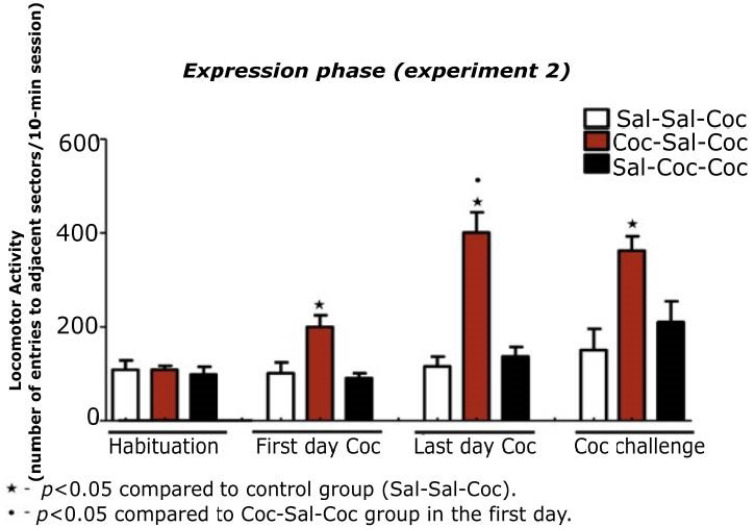
Locomotor activity of the animals in the expression phase during the habituation period, the first day of cocaine, the last day of cocaine, administered in a paired way (Coc-Sal-Coc) or not (Sal-Coc-Coc) to the environmental context, followed by the cocaine challenge after a not-handled 10-day period. The one-way ANOVA test was performed, followed by the Duncan post hoc test when necessary, and the GLM test for repeated measures to compare two measures of the same group at different times. Locomotor activity is expressed as mean ± SEM.

**Figure 5 biomedicines-11-00383-f005:**
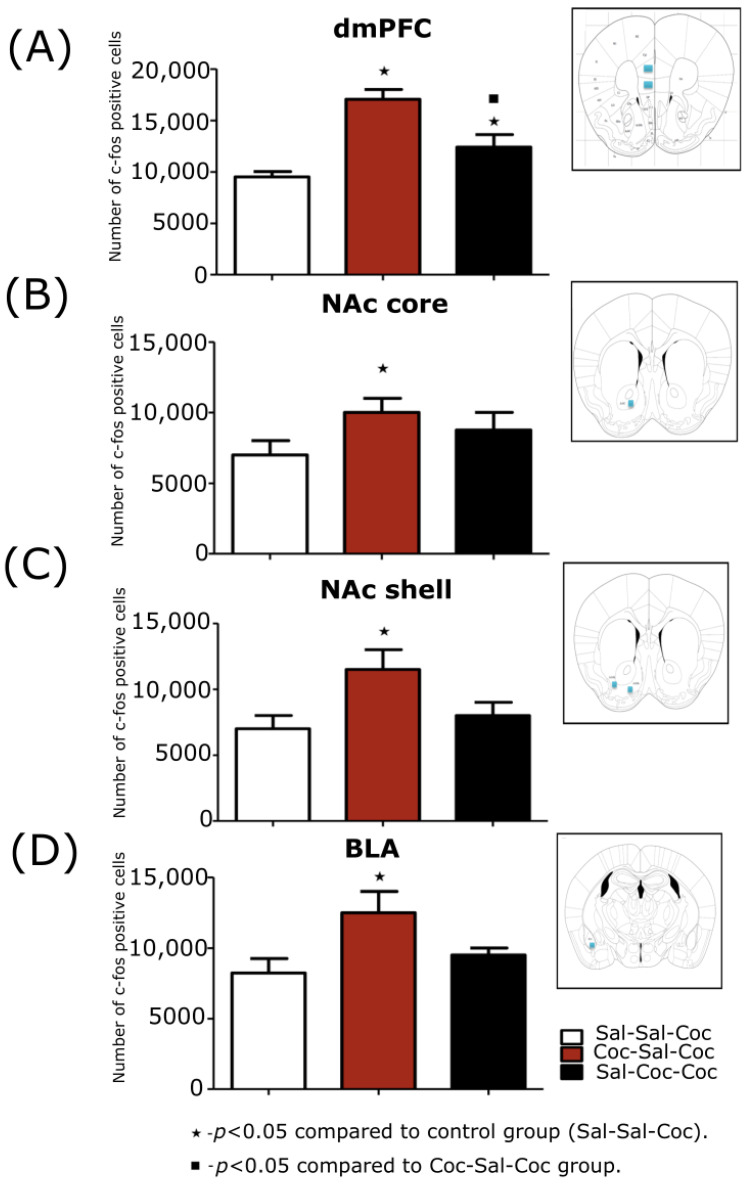
Stereological analysis of the expression c-Fos protein in animals treated with cocaine, administered in a paired way (Coc-Sal-Coc) or not (Sal-Coc-Coc) to the environmental context in the expression phase in the following structures: (**A**) The dorsomedial prefrontal cortex (dmPFC) measurements were performed at 1.95 mm of bregma, a representative area of quantification is observed by the blue rectangles on the right. (**B**) The nucleus accumbens core (NAc core) measurements were performed at 1.88 mm of bregma, a representative area of quantification is observed by the blue rectangle on the right. (**C**) The nucleus accumbens shell (NAc shell), measurements were performed at -1.18 mm of bregma, a representative area of quantification is observed by the blue rectangles on the right. (**D**) The basolateral amygdala (BLA), measurements were performed at -1.22 mm of bregma, a representative area of quantification is observed by the blue rectangle on the right. The number of c-Fos positive cells is expressed as mean ± SEM. The one-way ANOVA test was performed, followed by the Duncan post hoc test.

**Table 1 biomedicines-11-00383-t001:** Groups distribution according to the treatment received during the induction and expression phases.

Experiment 1 (Induction Phase)			
**Groups**	**Injection in the ** **open-field ** **(days 1, 3, 5, 7, 9, 11, 13, 15)** **Behavioral test ** **(days 1 and 15)**	**Injection in the ** **home-cage ** **(days 1, 3, 5, 7, 9, 11, 13, 15)** **Behavioral test ** **(days 1 and 15)**	
Sal-Sal	Saline	Saline	
Coc-Sal	Cocaine	Saline	
Sal-Coc	Saline	Cocaine	
Experiment 2 (Expression Phase)			
**Groups**	**Injection in the** **open-field ** (days 1, 3, 5, 7, 9, 11, 13, 15) **Behavioral test** (days 1 and 15)	**Injection in the ** **home-cage ** (days 1, 3, 5, 7, 9, 11, 13, 15) **Behavioral test** (days 1 and 15)	**Cocaine challenge and behavioral test in the open-field** (day 26)
Sal-Sal-Coc	Saline	Saline	Cocaine
Coc-Sal-Coc	Cocaine	Saline	Cocaine
Sal-Coc-Coc	Saline	Cocaine	Cocaine

**Table 2 biomedicines-11-00383-t002:** Stereological parameters determined for each structure.

Structure	Frame (µm × µm)	Grid (µm × µm)	Disector Height (µm)	Guard Zone (µm)
Dorsomedial prefrontal cortex	50 × 50	200 × 200	12	3
Nucleus accumbens core	100 × 67	175 × 175	12	3
Nucleus accumbens shell	100 × 67	175 × 175	12	3
Basolateral amygdala	98 × 58	150 × 150	12	3
Ventral tegmental area	60 × 60	150 × 150	12	3

**Table 3 biomedicines-11-00383-t003:** Comparative summary of c-Fos expression during both phases of the behavioral sensitization phenomenon.

Structures	Induction phase		Expression phase	
	Coc-Sal	Sal-Coc	Coc-Sal-Coc	Sal-Coc-Coc
dmPFC	↑	=	↑	↑↑
NAc core	↑	=	↑	=
NAc shell	=	=	↑	=
BLA	↑	=	↑	=
VTA	↑	↑↑	=	=

Abbreviations: dmPFC: dorsomedial prefrontal cortex; NAc core: nucleus accumbens core; NAc shell: nucleus accumbens shell; BLA: basolateral amygdala; VTA: ventral tegmental area. ↑: means increase expression of c-Fos compared to the control group; ↑↑: means increase in c-Fos expression compared to the Coc-Sal group of the Exp 1 and compared to the Coc-Sal-Coc group of the Exp 2; =: means the same amount of c-Fos expression for the 3 groups.

## Data Availability

The raw data associated with the findings of this study, including examples of images of IHC preparation used for the stereological counting, are publicly available to download from the link https://drive.google.com/drive/folders/1HJJUxdDpxlqOOf7KDfC_Dl9_y1g4ruxW?usp=sharing (accessed on 20 December 2022).

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
