# Peer review of "Distinctive Neuroanatomic Regions Involved in Cocaine-Induced Behavioral Sensitization in Mice"

_biomedicines, 2023, doi:10.3390/biomedicines11020383_

Round 1

Reviewer 1 Report

In the presented Article, the authors describe the results of their behavioural and immunohistochemical research in Swiss mice. The authors show the level of c-fos cells in the brain reward system loci of mice that underwent cocaine sensitization induction and cocaine sensitization expression protocols. The scientific approach presented in the manuscript is interesting, and the results are original.

My critical remarks concern the methodological elucidations, results presentation and the discussion of obtained results. The manuscript should be corrected and completed; therefore, my recommendation is MAJOR REVISION.

Specific Comments:

I)Since the environmental context of cocaine administration to induce and express locomotor sensitization to cocaine seem to be critically important in your data, please consider rephrasing the aim of your study to highlight the most crucial data you obtain.

II)Please improve methodological descriptions and Fig 1 layout to enable understanding of the experimental approach.

  1. Please indicate clearly (and justify the choice) the sex of mice that served in your study.
  2. It is unclear from the manuscript and the experimental design in Figure 1 what the schedule of cocaine administration is. Please provide precise information about cocaine dose in mg/kg and the plan for cocaine treatment. From the current version of the Article, the reader cannot understand how many daily cocaine injections in the „induction phase” were administered. Also, there is no clearly stated how many times from days 1-15 animals received cocaine injections in Experiments 1 and 2. Examples:
  3. Line 19: „second injection…… home cages for 15 days”. Do you mean that mice were injected twice, and the second injection was given somewhere within two week period?
  4. Line 116-117: „after 2 hours, the animals received a second injection of 10 mg/kg, i.p., of cocaine”. Do you mean that mice were injected twice daily during the „induction phase”
  5. Line 126: „During the alternate non-treatment days….”. Do you mean that animals received cocaine every second day: on days 1,3,5…., and 15?

  1. The treatment of all experimental groups generated in Experiment 1 and Experiment 2 needs to be clarified. Please provide clear information about the scheme of treatment for ALL six groups, in Experiment 1 for groups: „sal-sal”, „coc-sal”, „sal-coc”, and in Experiment 2 for groups: „sal-sal-coc”; „coc-sal-coc”; „sal-coc-coc”.
  2. Please remove redundancy from the experimental descriptions. If there were no differences in phases „Habituation” and „Induction”, please simplify explanations in 2.4.1 and 2.4.2 sections. Please consider also removing one panel from Fig 1. and indicate only the names of groups under the timeline, which were generated in Experiment 1 and in Experiment 2.
  3. The titles of 2.4.1 and 2.4.2 sections: „ Ethological and neuroanatomical….” do not reflect the content of these sections where drug treatment and environmental context of this treatment are indicated only. Please adjust titles to the content of these sections.
  4. The purpose of cocaine treatment in the open field/in-home cages is not clear from the descriptions in the manuscript, especially in the context of the generation of the model of locomotor sensitization to cocaine, not cocaine place preference. Please correct the explanations to understand the causal connection of the methodological approach to the modelling of the cocaine sensitization model. Please give a rationale for your experimental approach and add appropriate literature.
  5. There is no information on which way behavioural parameters were calculated. Please provide the necessary explanations to improve understanding of the Result description. Moreover, the description in chapter 2.3 suggests that authors measured in the open field apparatus more behavioural parameters than locomotor activity presented in the Results. If more results were performed in open field apparatus, they should all be described in the Result section. Please add lacking data or correct methodological descriptions.

III) Result section

  1. Based on Fig. 4 Cocaine challenge did not evoke locomotor sensitization in the sensitization expression phase vs sensitization induction phase. Please comment thoroughly in the discussion section. If necessary, tone down the title and the body text of the manuscript highlighting the story about cocaine sensitization effects
  2. How do you define the „frequency of locomotion” indicated as a Y-axis in Figure 2 and Figure 4? Please complete the description and correct axis title and add appropriate units.
  3. Comparing data obtained in experiments 1 and 2 is of critical importance. Please calculate data all together using factorial two-way or 3-way ANOVA. If it is impossible to combine experiments 1 and 2, please explain in detail the differences (and their meaning) you obtained in the induction phases of both investigations.
  4. Please provide statistical descriptions in Figure legends.

IV) Discussion section

  1. The limitation of the study is not indicated. Please provide
  2. Discussion of the environmental influence of cocaine administration to obtained results should be highlighted in the manuscript.

Minor comment:

Please correct the description in „Data Availability Statement” (lines 514-515) and provide information where interested readers can find raw data (e.g. with IHC pictures) which served for data preparation presented in manuscript.

Please correct the shortcut in line 38 „…addiction reduces natural rewards.”

Author Response

REPLIES TO REVIEWER #1

We thank the Reviewer for the comments and suggestions made and for carefully revising the text. All the revisions were corrected and included in the manuscript (highlighted in yellow).

In the presented Article, the authors describe the results of their behavioral and immunohistochemical research in Swiss mice. The authors show the level of c-fos cells in the brain reward system loci of mice that underwent cocaine sensitization induction and cocaine sensitization expression protocols. The scientific approach presented in the manuscript is interesting, and the results are original.

My critical remarks concern the methodological elucidations, results presentation, and the discussion of obtained results. The manuscript should be corrected and completed; therefore, my recommendation is MAJOR REVISION.

Specific Comments:

  1. Since the environmental context of cocaine administration to induce and express locomotor sensitization to cocaine seem to be critically important in your data, please consider rephrasing the aim of your study to highlight the most crucial data you obtain.

Response: We appreciate the suggestion. The aim was rephrased in the abstract and in the introduction (pages 1 and 2).

II)Please improve methodological descriptions and Fig 1 layout to enable understanding of the experimental approach.

  1. Please indicate clearly (and justify the choice) the sex of mice that served in your study.

Response: We thank the reviewer for his/her comments. Methodological descriptions and Fig 1 were improved in the new version of the manuscript.

We used female mice due to their high susceptibility to drugs, and to their robust and rapid behavioral response [1–4]. Moreover, the use of female mice positively contributes to a more ethical laboratory housing practice [5–7]. We also considered the translational aspects, since the number of women drug users is increasing significantly. Although some studies suggest that the greater vulnerability to drugs of abuse in females occurs due to hormonal variation [8], an increasing body of evidence suggests that there is no significant variability between female and male mice in several characteristics [9]. Another study that measured collected data from 40 different strains of mice in 3 different laboratories observed that female mice tested in randomized phases of the estrous cycle did not differ from males [10]. Furthermore, this same study showed that males themselves may present variability. Therefore, the variability in female behavior related to the hormonal cycle is not empirically established yet. We believe that, since we are measuring a pharmacological response to cocaine, we do not expect a great variability between subjects, allowing the use of females. Another important point is the stress caused by the vaginal smear when monitoring the estrous cycle. Several studies have shown that this procedure can alter the behavior of rodents because it increases corticosterone levels. Based on this, we used female mice without taking into account the estrous cycle, and the results generated in our experiments are closer to a real populational situation, and with more robust statistical data, since they represent a heterogeneous population.

  1. It is unclear from the manuscript and the experimental design in Figure 1 what the schedule of cocaine administration is. Please provide precise information about cocaine dose in mg/kg and the plan for cocaine treatment. From the current version of the Article, the reader cannot understand how many daily cocaine injections in the „induction phase” were administered. Also, there is no clearly stated how many times from days 1-15 animals received cocaine injections in Experiments 1 and 2. Examples:
  2. Line 19: „second injection…… home cages for 15 days”. Do you mean that mice were injected twice, and the second injection was given somewhere within two week period?
  3. Line 116-117: „after 2 hours, the animals received a second injection of 10 mg/kg, i.p., of cocaine”. Do you mean that mice were injected twice daily during the „induction phase”
  4. Line 126: „During the alternate non-treatment days….”. Do you mean that animals received cocaine every second day: on days 1,3,5…., and 15?

Response: We thank the Reviewer for these observations. The animals received saline or cocaine every second day for 15 days. It means they were injected on odd days 1,3,5,7,9,11,13 and 15 (total of 8 injections of cocaine and 8 injections of saline). During those eight days, they received a first injection, and 5 min later, they were tested in the open-field. Two hours later, they received a second injection in their home-cage. On the even days (2,4,6,8,10,12, and 14), animals were not manipulated. We included a new section and a table (page 3) explaining the groups, drug injection and local of the injection, and we included the details about cocaine dose and the schedule for cocaine treatment in the manuscript to clarify our experiment (page 4). Additionally, we improved the figure 1 including the details of the experimental design (page 5).

  1. The treatment of all experimental groups generated in Experiment 1 and Experiment 2 needs to be clarified. Please provide clear information about the scheme of treatment for ALL six groups, in Experiment 1 for groups: „sal-sal”, „coc-sal”, „sal-coc”, and in Experiment 2 for groups: „sal-sal-coc”; „coc-sal-coc”; „sal-coc-coc”.

Response: As already answered above, please see new section 2.2, where we explain in detail every group according to the experiment.

  1. Please remove redundancy from the experimental descriptions. If there were no differences in phases „Habituation” and „Induction”, please simplify explanations in 2.4.1 and 2.4.2 sections. Please consider also removing one panel from Fig 1. and indicate only the names of groups under the timeline, which were generated in Experiment 1 and in Experiment 2.

Response: We agree with the reviewer, and we clarified this part of the manuscript and removed the text with redundancy. Please see table 1 and figure 1.

  1. The titles of 2.4.1 and 2.4.2 sections: „ Ethological and neuroanatomical….” Do not reflect the content of these sections where drug treatment and environmental context of this treatment are indicated only. Please adjust titles to the content of these sections.

Response: We modify this part of the manuscript.

  1. The purpose of cocaine treatment in the open field/in-home cages is not clear from the descriptions in the manuscript, especially in the context of the generation of the model of locomotor sensitization to cocaine, not cocaine place preference. Please correct the explanations to understand the causal connection of the methodological approach to the modeling of the cocaine sensitization model. Please give a rationale for your experimental approach and add appropriate literature.

Response: We understand the Reviewer’s point and, as requested, we included a discussion about this issue in the manuscript (page 16). This protocol was thought to simulate a drug user with repeated use in a short time and after withdrawal that induced sensitization even with the same amount of exposure and time interval. The environmental context in which the user consumes drugs is relevant in the development of abusive use of drugs [11], and our study focused in characterizing which neuroanatomic structures relate to that behavior.

Considering that behavioral sensitization has two phases (induction and expression), our protocol was elaborated conceiving two groups with the same pharmacological history, one associated with the environmental context and the other not. Therefore, we sought to understand the interaction between behavioral sensitization and activation of neuroanatomical regions by observing the c-fos expression that reflects the association between the environmental context and cocaine administration.

  1. There is no information on which way behavioral parameters were calculated. Please provide the necessary explanations to improve understanding of the Result description. Moreover, the description in chapter 2.3 suggests that authors measured in the open field apparatus more behavioral parameters than locomotor activity presented in the Results. If more results were performed in open field apparatus, they should all be described in the Result section. Please add lacking data or correct methodological descriptions.

Response: The parameter chosen to measure the locomotor activity was the number of entries (frequency of entries) in the square with the four paws in the open-field arena. For each animal, the number of entries was evaluated in a 10-min session. It was considered a square only when the animal placed the four paws. Each animal generated a value that resulted from the sum of the entries, which in turn generated the mean for each group, and for statistical analysis all animals of all groups were considered. More details were included in section 2.4 (page 4). The details of this protocol have been double checked and there is no missing data, and no other parameter was analyzed, it was only the number of entries as it was described.

III) Result section

  1. Based on Fig. 4 Cocaine challenge did not evoke locomotor sensitization in the sensitization expression phase vs. sensitization induction phase. Please comment thoroughly in the discussion section. If necessary, tone down the title and the body text of the manuscript highlighting the story about cocaine sensitization effects.

Response: We thank the Reviewer for this suggestion, this part was clarified in the text. Please see lines 375 to 378, page 14.

  1. How do you define the „frequency of locomotion” indicated as a Y-axis in Figure 2 and Figure 4? Please complete the description and correct axis title and add appropriate units.

Response: We modified Y- the axis in figure 2 and 4 and the corresponding description of each figure.

  1. Comparing data obtained in experiments 1 and 2 is of critical importance. Please calculate data all together using factorial two-way or 3-way ANOVA. If it is impossible to combine experiments 1 and 2, please explain in detail the differences (and their meaning) you obtained in the induction phases of both investigations.

Response: Regarding the different protocols used in experiments 1 and 2, it will not be possible to compare them, since these data belong to independent groups. Animals in experiment 2 were exposed to cocaine challenge in order to evaluate the expression of the behavioral sensitization phenomenon, whereas animals from experiment 1 were not exposed to this challenge. Therefore, it will not be helpful to combine data as we are trying to evaluate two distinct phases.

  1. Please provide statistical descriptions in Figure legends.

Response: As required, we added the statistical description in the figure legends.

  1. IV) Discussion section
  2. The limitation of the study is not indicated. Please provide

Response: We thank for the comment. We added our work´s limitation on section 6 on page 17.

  1. Discussion of the environmental influence of cocaine administration on obtained results should be highlighted in the manuscript.

Response: We appreciate the suggestion. We added a discussion about the environmental influence of cocaine administration on lines 476-480, page 16.

Minor comment:

Please correct the description in „Data Availability Statement” (lines 514-515) and provide information where interested readers can find raw data (e.g., with IHC pictures), which served for data preparation presented in the manuscript.

Response: Our raw data are available in Data Availability Statement.

Please correct the shortcut in line 38 „…addiction reduces natural rewards.”

Response: We apologize for this mistake. The error was corrected in the manuscript.

References

[1]       Andersen SL, Teicher MH. Sex differences in dopamine receptors and their relevance to ADHD. Neurosci Biobehav Rev 2000;24:137–41. https://doi.org/10.1016/S0149-7634(99)00044-5.

[2]       Sneddon EA, Ramsey OR, Thomas A, Radke AK. Increased Responding for Alcohol and Resistance to Aversion in Female Mice. Alcohol Clin Exp Res 2020;44:1400–9. https://doi.org/10.1111/acer.14384.

[3]       Reichel CM, Chan CH, Ghee SM, See RE. Sex differences in escalation of methamphetamine self-administration: Cognitive and motivational consequences in rats. Psychopharmacology (Berl) 2012;223:371–80. https://doi.org/10.1007/s00213-012-2727-8.

[4]       Monroe S, Radke AK. Aversion-Resistant Fentanyl Self-Administration in Mice. Psychopharmacology (Berl) 2021;238:699–710. https://doi.org/10.1007/s00213-020-05722-6.Aversion-Resistant.

[5]       Caballero-Puntiverio M, Lerdrup LS, Arvastson L, Aznar S, Andreasen JT. ADHD medication and the inverted U-shaped curve: A pharmacological study in female mice performing the rodent Continuous Performance Test (rCPT). Prog Neuro-Psychopharmacology Biol Psychiatry 2020;99:109823. https://doi.org/10.1016/j.pnpbp.2019.109823.

[6]       Oliveira-Lima AJ, Marinho EAV, Santos-Baldaia R, Hollais AW, Baldaia MA, Talhati F, et al. Context-dependent efficacy of a counter-conditioning strategy with atypical neuroleptic drugs in mice previously sensitized to cocaine. Prog Neuro-Psychopharmacology Biol Psychiatry 2017;73:49–55. https://doi.org/10.1016/j.pnpbp.2016.10.004.

[7]       Clayton J, Collins F. NIH to balance sex in cell and animal studies. Nature 2014:282–3.

[8]       Hotsenpiller G, Horak BT, Wolf ME. Dissociation of conditioned locomotion and Fos induction in response to stimuli formerly paired with cocaine. Behav Neurosci 2002;116:634–45. https://doi.org/10.1037/0735-7044.116.4.634.

[9]       Prendergast BJ, Onishi KG, Zucker I. Female mice liberated for inclusion in neuroscience and biomedical research. Neurosci Biobehav Rev 2014;40:1–5. https://doi.org/10.1016/j.neubiorev.2014.01.001.

[10]     Mogil JS, Chanda ML. The case for the inclusion of female subjects in basic science studies of pain. Pain 2005;117:1–5. https://doi.org/10.1016/j.pain.2005.06.020.

[11]     Robinson TE, Berridge KC. The neural basis of drug craving: An incentive-sensitization theory of addiction. Brain Res Rev 1993;18:247–91. https://doi.org/10.1016/0165-0173(93)90013-P.

Reviewer 2 Report

1. Please provide in the methodology section the total number of animals used in the study.

2. Why did the Authors use female mice instead of male? Studies performed using female animals is quite problematic in the aspect of the results, as hormonal fluctuations are obvious. In line with this, is there any risk that the results obtained may be influenced by the hormone levels?

3. Please provide on what basis did the Authors choose the dose of cocaine (10 mg)? Were there any dose-response studies?

4. Some spelling mistakes can be observed, including punctuation errors.

Author Response

REPLIES TO REVIEWER #2

We thank the Reviewer for the comments and suggestions made and for carefully revising the text. All the revisions were corrected and included in the manuscript (highlighted in yellow).

  1. Please provide in the methodology section the total number of animals used in the study.

Response: We apologize for this missing information. The total number of animals used in our study was 91. It is now included in the revised version. Please see line 81, page 3.

  1. Why did the Authors use female mice instead of male? Studies performed using female animals is quite problematic in the aspect of the results, as hormonal fluctuations are obvious. In line with this, is there any risk that the results obtained may be influenced by the hormone levels?

Response: We used female mice due to their high susceptibility to drugs, and to their robust and rapid behavioral response [1–4]. Moreover, the use of female mice positively contributes to a more ethical laboratory housing practice [5–7]. Hormones can influence, although recent studies show that it is not only the hormones that influence these behaviors, and that males also present variability in several behaviors [9,10]. On the other hand, we have a more heterogeneous population, since we used female mice without distinguishing the estrous cycle, and we have a more faithful representation of the population, which strengthens the statistical analysis. Our results suggest that there is no need to submit the animals to the stressful procedure of vaginal smear. In addition, we also considered the translational aspects, females may be more sensitive to drugs and have a more exacerbated response, which may serve to alert the increase in drug use by women. This issue was included in the study limitation section on page 17.

  1. Please provide on what basis did the Authors choose the dose of cocaine (10 mg)? Were there any dose-response studies?

Response:  We thank the Reviewer for the observation. The literature shows that cocaine doses for the development of drug-seeking behavior range from 1 to 20 mg/kg [6,12–14]. Therefore, we chose an intermediate dose of 10 mg/kg. Additionally, previous studies conducted by our group [6,14,15] used the same dose as in the present manuscript.

  1. Some spelling mistakes can be observed, including punctuation errors.

Response: We thank the Reviewer for the suggestion. We improved and corrected the spelling and punctuation mistakes in the new version the manuscript.

References

[1]       Andersen SL, Teicher MH. Sex differences in dopamine receptors and their relevance to ADHD. Neurosci Biobehav Rev 2000;24:137–41. https://doi.org/10.1016/S0149-7634(99)00044-5.

[2]       Sneddon EA, Ramsey OR, Thomas A, Radke AK. Increased Responding for Alcohol and Resistance to Aversion in Female Mice. Alcohol Clin Exp Res 2020;44:1400–9. https://doi.org/10.1111/acer.14384.

[3]       Reichel CM, Chan CH, Ghee SM, See RE. Sex differences in escalation of methamphetamine self-administration: Cognitive and motivational consequences in rats. Psychopharmacology (Berl) 2012;223:371–80. https://doi.org/10.1007/s00213-012-2727-8.

[4]       Monroe S, Radke AK. Aversion-Resistant Fentanyl Self-Administration in Mice. Psychopharmacology (Berl) 2021;238:699–710. https://doi.org/10.1007/s00213-020-05722-6.Aversion-Resistant.

[5]       Caballero-Puntiverio M, Lerdrup LS, Arvastson L, Aznar S, Andreasen JT. ADHD medication and the inverted U-shaped curve: A pharmacological study in female mice performing the rodent Continuous Performance Test (rCPT). Prog Neuro-Psychopharmacology Biol Psychiatry 2020;99:109823. https://doi.org/10.1016/j.pnpbp.2019.109823.

[6]       Oliveira-Lima AJ, Marinho EAV, Santos-Baldaia R, Hollais AW, Baldaia MA, Talhati F, et al. Context-dependent efficacy of a counter-conditioning strategy with atypical neuroleptic drugs in mice previously sensitized to cocaine. Prog Neuro-Psychopharmacology Biol Psychiatry 2017;73:49–55. https://doi.org/10.1016/j.pnpbp.2016.10.004.

[7]       Clayton J, Collins F. NIH to balance sex in cell and animal studies. Nature 2014:282–3.

[8]       Hotsenpiller G, Horak BT, Wolf ME. Dissociation of conditioned locomotion and Fos induction in response to stimuli formerly paired with cocaine. Behav Neurosci 2002;116:634–45. https://doi.org/10.1037/0735-7044.116.4.634.

[9]       Prendergast BJ, Onishi KG, Zucker I. Female mice liberated for inclusion in neuroscience and biomedical research. Neurosci Biobehav Rev 2014;40:1–5. https://doi.org/10.1016/j.neubiorev.2014.01.001.

[10]     Mogil JS, Chanda ML. The case for the inclusion of female subjects in basic science studies of pain. Pain 2005;117:1–5. https://doi.org/10.1016/j.pain.2005.06.020.

[11]     Robinson TE, Berridge KC. The neural basis of drug craving: An incentive-sensitization theory of addiction. Brain Res Rev 1993;18:247–91. https://doi.org/10.1016/0165-0173(93)90013-P.

[12]     Wuo-Silva R, Fukushiro DF, Hollais AW, Santos-Baldaia R, Mári-Kawamoto E, Berro LF, et al. Modafinil induces rapid-onset behavioral sensitization and cross-sensitization with cocaine in mice: Implications for the addictive potential of modafinil. Front Pharmacol 2016;7:1–10. https://doi.org/10.3389/fphar.2016.00420.

[13]     Smith LN, Penrod RD, Taniguchi M, Cowan CW. Assessment of cocaine-induced behavioral sensitization and conditioned place preference in mice. J Vis Exp 2016;2016:1–13. https://doi.org/10.3791/53107.

[14]     Berro LF, Hollais AW, Patti CL, Fukushiro DF, Mári-Kawamoto E, Talhati F, et al. Sleep deprivation impairs the extinction of cocaine-induced environmental conditioning in mice. Pharmacol Biochem Behav 2014;124:13–8. https://doi.org/10.1016/j.pbb.2014.05.001.

[15]     Fukushiro DF, Carvalho R de C, Ricardo VP, Alvarez J do N, Ribeiro LTC, Frussa-Filho R. Haloperidol (but not ziprasidone) withdrawal potentiates sensitization to the hyperlocomotor effect of cocaine in mice. Brain Res Bull 2008;77:124–8. https://doi.org/10.1016/j.brainresbull.2008.05.004.

Reviewer 3 Report

In the present study, the authors characterized and compared the neuroanatomical structures involved in the induction and expression phases of the behavioral sensitization phenomenon by stereological quantification of c-Fos immunohistochemistry. To this end, in experiment 1 (induction phase), mice were treated with saline or cocaine (10 mg/kg) and challenged in the open-field apparatus. Then, these animals received a second injection of saline or cocaine in their home-cages for 15 days. In experiment 2 (expression phase), the protocol was followed, except after the conditioning period, the mice were not manipulated for 10 days. After this interval, all mice were challenged with cocaine (10 mg/kg). C-Fos staining was evaluated in the dorsomedial prefrontal cortex (dmPFC), nucleus accumbens core (NAc core and shell (NAc shell), basolateral amygdala (BLA), and ventral tegmental area (VTA). Neuroanatomical analysis indicated that in the induction phase, cocaine-conditioned animals had higher expression of c-Fos in the dmPFC, NAc shell, BLA, and VTA, whereas, in the expression phase, almost all areas had higher expression except for the VTA. Collectively, they suggest that environmental context plays a major role in the induction and expression of behavioral sensitization, although not all structures that compose the mesolimbic system contribute to this phenomenon.

The manuscript was well written and the experiment was well-designed with very good controls. I have some minor considerations.

1. Does both male and female mice used in the present study?

2. Line 440, typo error was found. Please remove the blank space between cocaine- and paired.

3. The authors discussed the dopamine and glutamate involved in the cocaine sensitization, the authors should also mention other neurotransmitters such as acid-sensing ion channels ( Jiang et al., Neuroscience, 246:170-8, 2013) responsible for cocaine sensitization.

Author Response

REPLIES TO REVIEWER #3

We thank the Reviewer for the comments and suggestions made and for carefully revising the text. All the revisions were corrected and included in the manuscript (highlighted in yellow).

In the present study, the authors characterized and compared the neuroanatomical structures involved in the induction and expression phases of the behavioral sensitization phenomenon by stereological quantification of c-Fos immunohistochemistry. To this end, in experiment 1 (induction phase), mice were treated with saline or cocaine (10 mg/kg) and challenged in the open-field apparatus. Then, these animals received a second injection of saline or cocaine in their home-cages for 15 days. In experiment 2 (expression phase), the protocol was followed, except after the conditioning period, the mice were not manipulated for 10 days. After this interval, all mice were challenged with cocaine (10 mg/kg). C-Fos staining was evaluated in the dorsomedial prefrontal cortex (dmPFC), nucleus accumbens core (NAc core and shell (NAc shell), basolateral amygdala (BLA), and ventral tegmental area (VTA). Neuroanatomical analysis indicated that in the induction phase, cocaine-conditioned animals had higher expression of c-Fos in the dmPFC, NAc shell, BLA, and VTA, whereas, in the expression phase, almost all areas had higher expression except for the VTA. Collectively, they suggest that environmental context plays a major role in the induction and expression of behavioral sensitization, although not all structures that compose the mesolimbic system contribute to this phenomenon.

The manuscript was well written and the experiment was well-designed with very good controls. I have some minor considerations.

  1. Does both male and female mice used in the present study?

Response: We thank for this question. No, only female mice were used in our work. We used female mice due to their high susceptibility to drugs, and to their robust and rapid behavioral response [1–4]. Moreover, the use of female mice positively contributes to a more ethical laboratory housing practice [5–7]. We also considered the translational aspects, since the number of women drug users is increasing significantly. Although some studies suggest that the greater vulnerability to drugs of abuse in females occurs due to hormonal variation [8], an increasing body of evidence suggests that there is no significant variability between female and male mice in several characteristics [9]. Another study that measured collected data from 40 different strains of mice in 3 different laboratories observed that female mice tested in randomized phases of the estrous cycle did not differ from males [10]. Furthermore, this same study showed that males themselves may present variability. Therefore, the variability in female behavior related to the hormonal cycle is not empirically established yet. We believe that, since we are measuring a pharmacological response to cocaine, we do not expect a great variability between subjects, allowing the use of females.

  1. In Line 440, typo error was found. Please remove the blank space between cocaine- and paired.

Response: The error was corrected in the manuscript, as well as some spelling and punctuation mistakes.

  1. The authors discussed the dopamine and glutamate involved in cocaine sensitization, the authors should also mention other neurotransmitters, such as acid-sensing ion channels ( Jiang et al., Neuroscience, 246:170-8, 2013) responsible for cocaine sensitization.

Response: We thank the Reviewer for the suggestion. We agree with the reviewer. This issue is important and a discussion about other neurotransmitters responsible for cocaine sensitization was added. We believe that the quality of the discussion is improved now. Please see lines 494-499, page 17.

References

[1]       Andersen SL, Teicher MH. Sex differences in dopamine receptors and their relevance to ADHD. Neurosci Biobehav Rev 2000;24:137–41. https://doi.org/10.1016/S0149-7634(99)00044-5.

[2]       Sneddon EA, Ramsey OR, Thomas A, Radke AK. Increased Responding for Alcohol and Resistance to Aversion in Female Mice. Alcohol Clin Exp Res 2020;44:1400–9. https://doi.org/10.1111/acer.14384.

[3]       Reichel CM, Chan CH, Ghee SM, See RE. Sex differences in escalation of methamphetamine self-administration: Cognitive and motivational consequences in rats. Psychopharmacology (Berl) 2012;223:371–80. https://doi.org/10.1007/s00213-012-2727-8.

[4]       Monroe S, Radke AK. Aversion-Resistant Fentanyl Self-Administration in Mice. Psychopharmacology (Berl) 2021;238:699–710. https://doi.org/10.1007/s00213-020-05722-6.Aversion-Resistant.

[5]       Caballero-Puntiverio M, Lerdrup LS, Arvastson L, Aznar S, Andreasen JT. ADHD medication and the inverted U-shaped curve: A pharmacological study in female mice performing the rodent Continuous Performance Test (rCPT). Prog Neuro-Psychopharmacology Biol Psychiatry 2020;99:109823. https://doi.org/10.1016/j.pnpbp.2019.109823.

[6]       Oliveira-Lima AJ, Marinho EAV, Santos-Baldaia R, Hollais AW, Baldaia MA, Talhati F, et al. Context-dependent efficacy of a counter-conditioning strategy with atypical neuroleptic drugs in mice previously sensitized to cocaine. Prog Neuro-Psychopharmacology Biol Psychiatry 2017;73:49–55. https://doi.org/10.1016/j.pnpbp.2016.10.004.

[7]       Clayton J, Collins F. NIH to balance sex in cell and animal studies. Nature 2014:282–3.

[8]       Hotsenpiller G, Horak BT, Wolf ME. Dissociation of conditioned locomotion and Fos induction in response to stimuli formerly paired with cocaine. Behav Neurosci 2002;116:634–45. https://doi.org/10.1037/0735-7044.116.4.634.

[9]       Prendergast BJ, Onishi KG, Zucker I. Female mice liberated for inclusion in neuroscience and biomedical research. Neurosci Biobehav Rev 2014;40:1–5. https://doi.org/10.1016/j.neubiorev.2014.01.001.

[10]     Mogil JS, Chanda ML. The case for the inclusion of female subjects in basic science studies of pain. Pain 2005;117:1–5. https://doi.org/10.1016/j.pain.2005.06.020.

Round 2

Reviewer 1 Report

Minor comments:

1. In the 2.4 section (line 107), please add dimensions of „19 squares”.

2. Please rephrase the Y-axis title in Fig. 2 and Fig. 4 to understand the calculation method of numbers indicated in the Y-axis. Would < number of entries to adjacent square> be more effective? Alternatively, based on square dimension (see my previous comment), you could express the locomotor activity data as distance travel in [cm] and then, in an easy way, describe locomotor activity changes in the Result section and the graphs.

3. Please add units to column „Frame” in Table 2.

Author Response

We thank the Reviewer for the suggestions and for the carefully reading of the manuscript. The manuscript was reviewed, and all the suggestions and points raised by the Reviewer were included in the new version.

  1. In the 2.4 section (line 107), please add dimensions of „19 squares”.

Response: To improve the details of the open-field arena, we included more information about the apparatus. We also replaced “square” for “trapezoid-shape sector” to be more precise, since the sectors are not square-shape. The apparatus consisted of an open top cylinder with a circular plastic wall (40 cm in diameter and 50 cm high), with a floor divided into 19 similar trapezoid-shape sectors of approximately 67.51 cm2 each, delimited by three concentric circles of different radii (8, 14, and 20 cm) intersected by radial line segments (Wuo et al, 2019).

  1. Please rephrase the Y-axis title in Fig. 2 and Fig. 4 to understand the calculation method of numbers indicated in the Y-axis. Would < number of entries to adjacent square> be more effective? Alternatively, based on square dimension (see my previous comment), you could express the locomotor activity data as distance travel in [cm] and then, in an easy way, describe locomotor activity changes in the Result section and the graphs.

Response: We rephrased the Y-axis title in Fig. 2 and Fig. 4 to “number of entries to adjacent sectors/10-min session” The measurements we obtained were by manual counter (see study limitation section), and to measure the distance traveled accurately, it would be necessary to use a special software and camera. Moreover, as it is a circular open field, the floor is a trapezoid-shape sectors of different areas not uniformly divided, and the measurement of distance based on a calculation wouldn´t be comparable.

  1. Please add units to column „Frame” in Table 2.

Response: We apologize for this missing information. As suggested, the unit (µm x µm) was included in Frame column of Table 2.